# Efficient Low-Frequency SSVEP Detection with Wearable EEG Using Normalized Canonical Correlation Analysis

**DOI:** 10.3390/s22249803

**Published:** 2022-12-14

**Authors:** Victor Javier Kartsch, Velu Prabhakar Kumaravel, Simone Benatti, Giorgio Vallortigara, Luca Benini, Elisabetta Farella, Marco Buiatti

**Affiliations:** 1Department of Electrical, Electronic, and Information Engineering “Guglielmo Marconi”, University of Bologna, 40136 Bologna, Italy; 2Digital Society Center, Fondazione Bruno Kessler, 38123 Trento, Italy; 3Center for Mind/Brain Sciences (CIMeC), University of Trento, 38068 Rovereto, Italy; 4Department of Sciences and Methods for Engineering, University of Modena and Reggio Emilia, 41123 Reggio Emilia, Italy; 5Department of Information Technology and Electrical Engineering at ETH Zurich, 8092 Zurich, Switzerland

**Keywords:** SSVEP, CCA, NCCA, wearable EEG, frequency tagging, delta band, theta band

## Abstract

Recent studies show that the integrity of core perceptual and cognitive functions may be tested in a short time with Steady-State Visual Evoked Potentials (SSVEP) with low stimulation frequencies, between 1 and 10 Hz. Wearable EEG systems provide unique opportunities to test these brain functions on diverse populations in out-of-the-lab conditions. However, they also pose significant challenges as the number of EEG channels is typically limited, and the recording conditions might induce high noise levels, particularly for low frequencies. Here we tested the performance of Normalized Canonical Correlation Analysis (NCCA), a frequency-normalized version of CCA, to quantify SSVEP from wearable EEG data with stimulation frequencies ranging from 1 to 10 Hz. We validated NCCA on data collected with an 8-channel wearable wireless EEG system based on BioWolf, a compact, ultra-light, ultra-low-power recording platform. The results show that NCCA correctly and rapidly detects SSVEP at the stimulation frequency within a few cycles of stimulation, even at the lowest frequency (4 s recordings are sufficient for a stimulation frequency of 1 Hz), outperforming a state-of-the-art normalized power spectral measure. Importantly, no preliminary artifact correction or channel selection was required. Potential applications of these results to research and clinical studies are discussed.

## 1. Introduction

Electroencephalography (EEG) is a versatile, non-invasive, low-cost neuroimaging technique that records brain activity with an excellent temporal resolution to investigate brain function and pathology [1]. In general, extracting functional brain responses from EEG signals requires long-lasting, repeated stimulus presentations because of the interference of endogenous EEG activity and artifacts of biological and technical origin. One successful experimental paradigm (hereby termed “frequency tagging”) that minimizes this constraint is based on strictly periodic stimulation and exploits the property of the brain activity to respond to a sensory stimulus presented periodically at a specific (“tag”) temporal frequency by resonating at the same frequency during the stimulation period [2,3].

This effect is manifested in the EEG recordings by a sharp peak in the signal’s power spectrum at that specific “tag” frequency. Since the ongoing EEG activity is broadband in frequency, the stimulus-related response in the frequency domain is very easily discriminated from the stimulus-unrelated activity, yielding a much higher Signal-to-Noise Ratio (SNR) than the one obtained with the repetitive presentation of single, temporally isolated stimuli (event-related paradigm). Moreover, since most EEG artifacts (eye movements, blinks) are also broadband in frequency, frequency tagging is more robust to artifacts and requires a lighter artifact rejection procedure than event-related designs.

Frequency tagging is widely used to test the integrity of sensory processing, especially in the visual domain (Steady-State Visually Evoked Potentials—SSVEP [3]) and in the auditory domain (Auditory Steady-State Responses—ASSR [2]). Typical presentation rates depend on the frequency ranges in which these sensory systems are most responsive: 8–20 Hz for SSVEP and around 40 Hz for ASSR. In these frequency ranges, it is possible to obtain very high SNR because the amplitude of both ongoing EEG activity and biological EEG artifacts is low.

Recent studies have extended the use of frequency tagging to lower frequencies (0.5–7 Hz, encompassing the classical delta and theta frequency ranges) either because they focused on higher-level neural processing characterized by longer temporal scales (e.g., syllables and words in speech perception in adults [4] and infants [5]), or because the experimental design was based on the infrequent presentation of key stimuli among control ones (e.g., selectivity for faces among other visual objects in adults [6] and infants [7]), or because neural processing is slow due to the immaturity of the visual system (e.g., face perception in newborns [8]). Obtaining reliable brain responses within this low-frequency range is more challenging because since frequencies are lower, longer presentations are needed to capture the related oscillatory responses, and the interference of both EEG ongoing activity and artifacts increases for decreasing frequencies. Still, even at low frequencies, the spectral specificity of the stimulus-related response and the robustness to artifacts of frequency tagging makes it very well-suited for neuro-cognitive testing of special populations with limited attention span, such as infants and patients [9], and/or for out-of-the-lab applications.

Recent technological advances in microprocessor design, wireless communication, and innovative dry electrodes prompted the development of a new generation of wearable wireless EEG systems that allow recording subjects’ brain activity outside controlled laboratory settings [10,11]. Several studies have shown that, despite the low number of channels and potential risk of higher noise, these portable EEG systems may be used with success to rapidly and reliably record SSVEP and ASSR within their optimal frequency range (8–40 Hz) for clinical testing and BCI systems [12,13].

On the algorithmic side, steady-state evoked potentials are traditionally computed from the power spectrum at the tag frequency, often normalized by the power spectrum at neighboring frequency bins [3,4,8]. Another up-and-coming technique, very successful in the field of SSVEP-based BCI systems, is Canonical Correlation Analysis (CCA) [14,15,16]. CCA presents appealing features for processing SSVEP signals: it potentially provides infinite frequency resolution; as an intrinsically multivariate analysis, it does not require an explicit channel selection strategy; it can correlate EEG with several harmonics (together with central frequency) while conveying the information in a single output.

However, it is currently unclear which is the best analysis tool for fast and reliable detection of SSVEP from low-density, wearable, wireless EEG recordings and stimulation in the low-frequency range used for higher-level brain responses and/or with infants and newborns (0.5–7 Hz), and what is the minimum data length necessary to obtain such a reliable response.

For this purpose, we recorded SSVEP responses of 15 adult subjects presented with sinusoidal on-off checkerboard visual stimulation at four different frequencies, ranging from values typically used in BCI applications (7–10 Hz) [10,17] to values typical of high-level cognition and/or neural processing in newborns and infants (1–3 Hz) [4,5,8]. To better quantify the “peak” of the brain response at the stimulation frequency with respect to the wide fluctuations in the background EEG at low frequencies, we tested a frequency-normalized version of CCA (NCCA [18,19]) and compared it with the more traditional normalized power spectrum [3] for the detection of the SSVEP response. To tackle the issue of the limited amount of data in special populations, such as newborns, infants, and patients, we systematically investigated, for each stimulation frequency, the performance of the two methods for progressively shorter time windows to estimate the minimum length of recording necessary to extract a reliable SSVEP.

The paper is organized as follows. Section 2 describes the EEG setup, the experimental paradigm and the analysis methods implemented for this study. Section 3 comparatively illustrates the performance of NCCA and normalized power spectrum on the recorded data as a function of data length. Section 4 provides a discussion about the results and their potential applications. Finally, Section 5 draws the conclusions of this work.

## 2. Material and Methods

### 2.1. Acquisition System

BioWolf is a wearable medical-grade BCI platform enabling wireless raw data acquisition, and onboard digital signal processing [10]. The platform is built around the ADS1298 (Texas Instruments), often regarded as a de facto standard device for embedded biopotential acquisition (favorable trade-off between signal quality and energy efficiency), which allows simultaneous sampling of up to eight bipolar channels (24 bits) at a maximum sample rate of 32 kbps.

For onboard processing, BioWolf integrates Mr Wolf [20], an energy-efficient Parallel Ultra-Low-Power (PULP) System on Chip (SoC), which enables efficient real-time execution of complex algorithms such as the ones required for biosignal-based BCIs.

Additionally, the platform integrates an Arm Cortex-M4 SoC from Nordic (nRF52832), acting as a wireless transmission device (BLE 5.0) for raw data or computation outputs, an electrode contract quality subsystem, a battery manager (BQ25570), and an IMU device (IIS2DH) to register acceleration and temperature. The software environment built around the platform allows real-time data visualization and storage and up to 256 digital triggers that can be used to synchronize the raw data to a given event.

In this work, BioWolf is coupled with a custom elastic EEG cap featuring eight dry soft electrodes from Idun Technologies [21], mapped to O2,Oz,O1,P6,P2,P1,P5, and CPZ (following the 10–20 reference system). Each electrode comprises an active circuitry [22] to enhance the overall CMRR and reduce artifacts due to cable movement. Figure 1 offers a graphical summary of the complete system.

### 2.2. Experimental Protocol

Fifteen healthy subjects (aged 25–35 years, 1 female and 14 males) participated in data collection, performed in a quiet office environment with dimmed light and away from well-known sources of electrical interference. All subjects reported no neurological or psychiatric disorders and gave their informed consent for inclusion before they participated in the study. The study was conducted in accordance with the Declaration of Helsinki, and the protocol was approved by the Ethics Committee of the University of Trento (Protocol 2019-042).

Stimuli consisted of a sinusoidal on-off 100% contrast temporal modulation black and white 10 × 10 square checkerboard, presented on a uniform grey background at a distance of 80 cm from the subject’s eyes and subtending a visual angle of approximately 15 × 15° [23]. A sinusoidal contrast modulation was used because it generates fewer harmonics [3] and because, since the on-off dynamics are smooth, it is a more pleasant and less fatiguing visual stimulation than a squarewave stimulation mode for the subjects. The stimuli also included a thin black diagonal cross to help visual fixation. We used four stimulation frequencies in the interval between 1 and 10 Hz. Since each stimulation frequency was used not only to detect the response at that specific frequency and its harmonics but also as a control when testing the detection of other frequencies, in order to avoid the interference of the harmonics of lower frequencies with higher frequencies, we selected four incommensurable stimulation rates: 1, 3.125, 7.8125, and 10.6125 Hz. Three trials per frequency (single trial duration = 25 s) were presented in a randomized order. A minimum of 10 s inter-trial rest was also included to reduce visual fatigue. Textures were generated with Psychtoolbox 3.0.12 based on Matlab 2020b (MathWorks Inc., Natick, MA, USA) for Windows.

Before each experimental session, hardware (battery and triggers) and electrode contact quality checks were performed. Electrode quality was adjusted using conductive gel when required.

### 2.3. Canonical Correlation Analysis

CCA computes the linear dependency between two multidimensional variables by finding a couple of linear combinations, maximizing their correlation, one for each multidimensional variable. As per its robustness, CCA recently gained significant popularity in SSVEP-based BCI applications [14].

Specifically, CCA retrieves a set of maximized correlations (canonical coefficients), each resulting from a couple of linear combinations belonging to subspaces that are orthogonal to each other. In an SSVEP context, the two multidimensional variables are the *n* EEG input channels and a set of *m* reference signals that identify the frequency of one single stimulus, usually the sine and cosine of the frequency in an exam and one or more harmonics.

Our CCA implementation relies on the Golub algorithm [24], which performs a QR decomposition step over the input and the reference signals, followed by an SVD factorization of the product between the two orthogonal matrices. The Euclidean Norm is then applied over the resulting canonical coefficients (size d=min(n,m)) to provide a single correlation value.

Given the intrinsic noisy nature of EEG signals collected with portable devices such as BioWolf [12], signals fed to the CCA core are first filtered. In this work, we applied a 10-tap IIR high-pass filter (Cut: 0.2 Hz, Pass: 0.4 Hz). No other preprocessing signal step (such as artifact correction/rejection) was performed. Notably, preliminary tests showed that the artifact removal preprocessing used for FTA (see Section 2.4) does not improve CCA performance.

Figure 2 provides a block diagram of the CCA algorithm, including the preprocessing steps.

#### 2.3.1. Normalized Canonical Correlation Analysis (NCCA)

NCCA builds on top of traditional CCA, aiming to provide an index of the presence of a “peak” at a given frequency. NCCA is obtained by computing the ratio between the CCA response for a central (tag) frequency $Corr_{cf}$ and the mean value (often called background) between two adjacent frequencies $Corr_{cf+\Delta }$ and $Corr_{cf-\Delta }$, as denoted in the following:(1)$${\mathrm{NCCA}}_{tf}=\frac{Corr_{cf}}{mean(\left[Corr_{cf+\Delta }Corr_{cf-\Delta }\right])}$$

In this work, NCCA is computed on a single segment length for each trial (typically three trials per stimulation frequency). The trial results are then averaged to obtain a single CCA value per subject. The background width of 0.4 Hz (Δ 0.2Hz from $Corr_{cf}$) was used for NCCA values reported in this work, empirically found to provide the best performance.

#### 2.3.2. CCA-Based Frequency Response Computation

For illustrative and comparison purposes (as it will be introduced in Section 3), CCA was also employed to perform a frequency spectrum estimation (commonly achieved through the Fourier Transform). For this, CCA is computed for a range of frequencies (the range of interest). The start of the frequency range and the ΔHz increments have been selected to precisely hit a tag frequency (typically at the center of the frequency range). As in the rest of this work, this estimation also takes advantage of the information in the signal harmonics.

### 2.4. Frequency Tagging Analysis (FTA)

Frequency Tagging Analysis (FTA) is a simple and effective technique based on Fourier analysis to estimate the SSVEP response. FTA computes the frequency-specific neural response for each tag frequency by first performing a Fast Fourier Transform (*FFT* function, MATLAB) that precisely includes a frequency bin coinciding with the tag frequency by using a window length corresponding to a finite number of stimulation cycles. In this work, as a trade-off between the frequency resolution and the shortest window used, we set the window length to 4 s for 1 Hz stimulation (i.e., 4 cycles), 2.56 s for 3.125 Hz (i.e., 8 cycles), 1.536 s for 7.8125 Hz (i.e., 12 cycles), and 1.5 s for 10.6125 Hz (i.e., 16 cycles). For each stimulation frequency, EEG data from each trial were segmented in half-overlapping epochs of the corresponding window length, and the power spectrum of each electrode was computed as follows:(2)$$PS\left(f\right)={\langle F\left(f\right)\times F^{*}\left(f\right)\rangle }_{ep}$$
where *F(f)* is the Fast Fourier transform, and the average is computed across all the epochs belonging to a specific stimulation frequency. Since preliminary tests showed that there is no benefit in channel selection, for subsequent analyses, the power spectrum was averaged across all electrodes.

As for CCA processing, FTA requires a preliminary filtering step: the raw data are first low-pass filtered at a cut-off frequency of 40 Hz (default EEGLAB filter [25,26]) followed by a non-causal high-pass filter between 0.15 and 0.3 Hz with a stop-band attenuation of 80 dB.

Additionally, before FTA processing, data are preprocessed using the Artifacts Subspace Reconstruction (ASR) algorithm [27] (as we observed poor performance in SSVEP detection using the raw filtered data). For this, the default settings (*k* = 20, ASR Removal) are used, as they empirically yield better results [12,28,29]. Trials with a duration of less than 20 s were discarded from further analyses.

#### Frequency-Tagged Response (FTR)

To quantify the SSVEP amplitude at the stimulation frequencies, we used the Frequency-Tagged Response (FTR) proposed in [8]. FTR is defined as the ratio between the power spectrum at the tag frequency and the power-law fit of the background power spectrum at the tag frequency estimated by fitting a line (MATLAB function *polyfit*) to the logarithm of the power at the four neighboring frequency bins (two from each side of the tag frequency bin).

### 2.5. Statistical Analysis and Sample Size

For both NCCA and FTA, we computed the statistical significance of the peak (the SSVEP response compared to the background EEG) by performing a one-tailed Wilcoxon signed rank test [30] between the CCA/power spectrum at the tag frequency and the average CCA/power spectrum at the neighboring frequencies used to estimate the background EEG in the NCCA/FTR computations. We used the Wilcoxon signed rank test because it is a non-parametric test [30] that does not require the assumption of normality, which, in the case of EEG data, is usually not guaranteed.

To determine the minimal sample size required to reliably observe the hypothesized effects, we used an independent dataset (11 adult subjects) recorded with another EEG system (64 channels, Brain Products, Munich, Germany) with visual stimulation (black and white reversing checkerboards) and stimulation frequency range very similar to the ones used in this work. For more details about the EEG system, data preprocessing, and visual stimulation, see [23], where part of this dataset has been analyzed. We computed the effect size of the FTA (the standard measure used in this work) from the difference between the power spectrum at the tag frequency and the estimated background power at the tag frequency for a window of 20 s of data and a stimulation frequency of 4 Hz (i.e., in the lower, noisier part of the frequency range used in this work) on electrode POz of the independent dataset, which is located in the middle of the set of electrodes used in this work. The effect size was dz=1.30. The minimal sample size, calculated with the software G*Power [31] (a priori power analysis, Wilcoxon signed-rank test, matched pairs, one tail, and α=0.05), was 9. Therefore, the number of subjects (n=15) used in this work is abundantly sufficient to observe the expected effect size.

## 3. Results

### 3.1. Both CCA and Power Spectrum Detect SSVEP Responses with “Long” Data Segments

For long recording segments (20 s), both CCA and power spectrum show a clear SSVEP response at each visual stimulation in the form of a peak at the stimulation frequency (Figure 3 showing CCA (top row) and power spectrum (bottom row) at the four stimulation frequencies and, for a control comparison, at rest, for a representative subject). However, the two measures differ in the peak’s sharpness and fluctuations of the resting state EEG. Especially for the two lower stimulation frequencies (1 and 3.125 Hz), the power spectrum of the rest condition shows wide fluctuations; this effect combined with the low-frequency resolution of the power spectrum (0.25 Hz for 1 Hz tag frequency and 0.4 Hz for 3.125 Hz tag frequency) due to the trade-off between the number of cycles and window length (see Section 2.4) and the 1/f neural noise leads to the potential spurious detection of peaks in the rest condition. On the contrary, CCA, which has the advantage of a virtually infinite frequency resolution, shows a sharp peak at all frequencies, even the lowest one, and a much smoother and flatter profile for the rest condition.

### 3.2. NCCA vs. FTR Detection Performance in Function of Window Length

We then tested the performance of both NCCA and FTR in detecting the SSVEP response at the stimulation frequency for progressively shorter recording time windows. The shortest window is 2 s for NCCA and 4 s for FTR (as the minimal window for detecting the peak at 1 Hz is 4 s, see Section 2.4). Figure 4 shows the values reported by NCCA and FTR computed at each one of the stimulation frequencies on the segments corresponding to the four stimulation frequencies and on the rest segments, averaged over all subjects. For increasing window length, NCCA rapidly departs from 1 (the value corresponding to no SSVEP response) specifically for the detector of the stimulation frequency, while its measures on the segments relative to stimulation at the other three frequencies, as well as on the rest segments, never increase and keep fluctuating around 1. This effect is clear for the NCCA at the three higher frequencies even from the shortest time window (2 s), while for the most challenging frequency (1 Hz), the effect starts to be evident from the 4 s time window.

FTR detectors are also successful in detecting the correct SSVEP response, yet they show a lower specificity, in particular for the lowest frequency (1 Hz): while FTR(1Hz) is higher than 1 for 1 Hz stimulation frequency even at the lowest time window (4 s), it is higher than one also for the other stimulation frequencies and for rest segments. FTR detectors for higher frequencies better discriminate the stimulation frequency, but the 10 Hz one also slightly rises on the resting segments, probably due to some difficulty in distinguishing the SSVEP response from the spontaneous alpha peak.

These results are confirmed by testing the statistical significance of the peaks at each stimulation frequency (one-tailed Wilcoxon signed rank test between the CCA/power spectrum at the stimulation frequency and the CCA/power spectrum of the background EEG estimated from the neighboring frequency bins, see Methods): for 1 Hz, the statistical value of the CCA peak crosses the threshold of p=0.01 already from 4 s windows and rapidly reaches its maximum at 8 s, while one of the power spectrum peaks crosses the same threshold only at 6 s and increases very slowly for longer time windows. For 3.125 Hz, while both measures show high statistical significance from the shortest window, the CCA peak is already highly significant (p≪0.01) with 2 s of data, while FTR statistics improve much more slowly with the window length. This suggests that for a reliable SSVEP neural response at low stimulation frequencies, NCCA requires shorter data than the conventional FTA. Statistical significance for the detectors at the two higher frequencies is similar for the two measures, but NCCA is always at a plateau while FTR is occasionally lower.

## 4. Discussion

Due to their reliability and high SNR, SSVEP are widely used in a variety of experimental protocols, both in neuroscience research [3] and in clinical applications, such as BCI [32]. The overwhelming majority of these studies use stimulation frequencies higher than 5–6 Hz [3,33] because they are based on the evoked potentials generated by the early visual system, which responds much more poorly at lower frequencies [34]. Another line of research showed that SSVEP at lower frequencies (0.5–6 Hz) might be used to track the neural responses associated with higher-order processing, such as face perception [6,7,8], spatial relations between social entities [35], biological movement [36], and reading [37], both in adults and through development [9]. However, all these latter studies recorded the data with non-portable high-density EEG systems in controlled laboratory settings.

Here, for the first time, we demonstrate that it is possible to reliably detect SSVEP with a wearable, fully portable, wireless EEG with stimulation frequencies in the delta and theta frequency range down to 1 Hz in as rapidly as 4 s. To obtain this result, we tested NCCA, a frequency-normalized version of CCA, on data collected with a wearable, wireless EEG system based on BioWolf, a compact, ultra-light, ultra-low power recording platform [10], with four stimulation frequencies ranging from 1 to 10 Hz, and compared its performance with the traditional Fourier-based power spectral analysis (FTA).

We first observed that with a reasonable amount of data (20 s), CCA shows a clear, sharp peak over a shallow background for all the stimulation frequencies; the peak of the power spectrum is equally sharp for the two highest frequencies only, while it is increasingly lower and comparable to the background power fluctuations for the two lowest frequencies (Figure 3). These observations anticipate the performance of NCCA, which shows high specificity for the stimulation frequency (Figure 4) and high statistical significance of peak detection (Figure 5) at all stimulation frequencies, with data windows as short as 2 s for the three highest frequencies, and as short as 4 s for the lowest frequency. While FTR displays similar results for the two highest frequencies, and to some extent for 3.125 Hz, at 1 Hz it fails to discriminate between different stimulation frequencies and has far lower statistical power than NCCA.

The reasons for the higher performance of NCCA compared to FTA likely rely on some key differences between the two methods that are particularly relevant for the low-frequency range analyzed in this work:

(1) Frequency resolution: Since FTA’s frequency resolution is inversely proportional to the window length used to compute the FFT, it drastically decreases for decreasing frequencies, which is detrimental for the detection of a frequency peak because the sampling of the background EEG becomes poor; on the other hand, CCA frequency resolution is unbounded, as it only depends on reference signals, which can be generated at any specific frequency. This constraint in the choice of the frequency bins, together with the fact that the background EEG fluctuations increase with decreasing frequencies (1/f phenomenon), cause the FTR detector for 1 Hz to be much less efficient than the NCCA one, as it is evident from the left-hand panels of Figure 3.

(2) Channel selection: Since FTA is computed for each channel, merging the frequency response information across channels is not a trivial task, as less responsive channels may heavily smear the average response; on the other hand, CCA is a multivariable statistical method that optimally selects the relevant information from all the channels and provides a single response without requiring any preliminary channel selection.

(3) Response at harmonic frequencies: CCA effortlessly incorporates the information in the tag frequency and its harmonics by using multiple reference signals, while FTA needs to be computed for each specific frequency, and averaging across harmonics may underestimate the overall response.

(4) Artifact removal: Especially in the low-frequency range where artifacts are more relevant, FTA requires preliminary artifact removal, as previously shown on similar data [12] and confirmed by directly applying FTA to the raw data of this work (see Section 2.4 for details on preprocessing), which causes a reduction in data statistics; in contrast, CCA is computed on all the available data as it does not require any artifact preprocessing (preliminary artifact removal does not improve CCA performance, see Section 2.3).

The estimates of the minimal data length required for SSVEP detection identified in this work have far-reaching practical consequences on the application of frequency tagging paradigms to special populations, such as newborns, infants, and patients that typically have a limited attention span [9]. For example, this means that we may be able to record a neural response associated with face perception in newborns (requiring the contrast between two different stimulations [8]) from as short as two EEG data segments of 7–8 s each, during which newborns are attentive, a perspective suggesting potential implications for the development of related biomarkers to be recorded in out-of-the-lab scenarios.

These results also demonstrate that low-density wearable EEG devices based on platforms such as BioWolf [10,12], typically used in a BCI context with high-stimulation frequencies (8–20 Hz) [16,38,39], can also provide reliable signal quality even with very low-stimulation frequencies (1 Hz). This denotes an excellent prospect for using these devices in research and clinical studies on various neurocognitive functions in out-of-the-lab environments and/or with special populations, such as newborns, infants, or patients.

Further research is needed to test the efficiency of NCCA in developing or aging populations, which might challenge its robustness to artifacts. Further, exploring other stimulation modalities would test the possibility of extending it to other fields, such as speech processing [4,5] or pain processing [40].

We will publicly share a user-friendly version of NCCA as open-source software in the near future.

## 5. Conclusions

In this work, we presented NCCA, a measure based on a frequency-normalized version of CCA which efficiently detects SSVEP at very low-frequencies (1–6 Hz) without any artifact preprocessing on EEG data recorded with a low-density, ultra-light wearable EEG device. We further showed that the proposed detection method requires extremely short data segments for a reliable SSVEP estimation, even for the lowest stimulation frequency tested (1 Hz). These results pave the way to efficient out-of-the-lab neurocognitive testing in special populations, such as newborns, infants and patients.

## Figures and Tables

**Figure 1 sensors-22-09803-f001:**
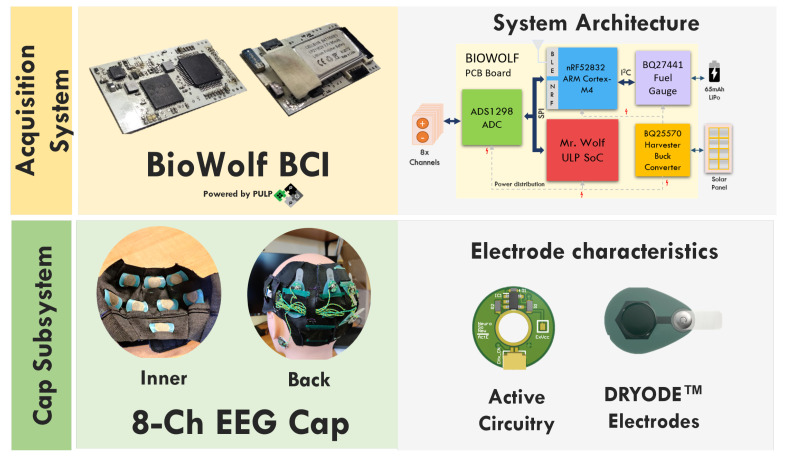
Portable EEG system embedding BioWolf and eight Dryode^TM^ electrodes featuring active signal buffering.

**Figure 2 sensors-22-09803-f002:**
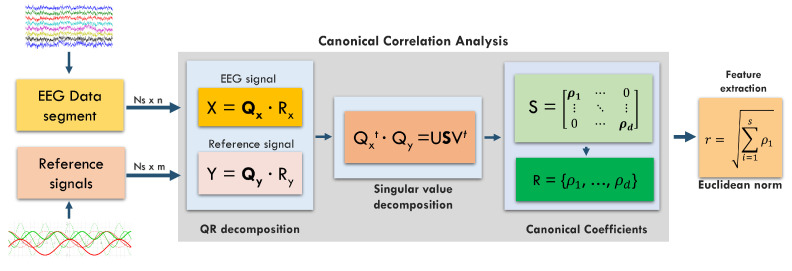
Block diagram for the CCA computation, including preprocessing.

**Figure 3 sensors-22-09803-f003:**
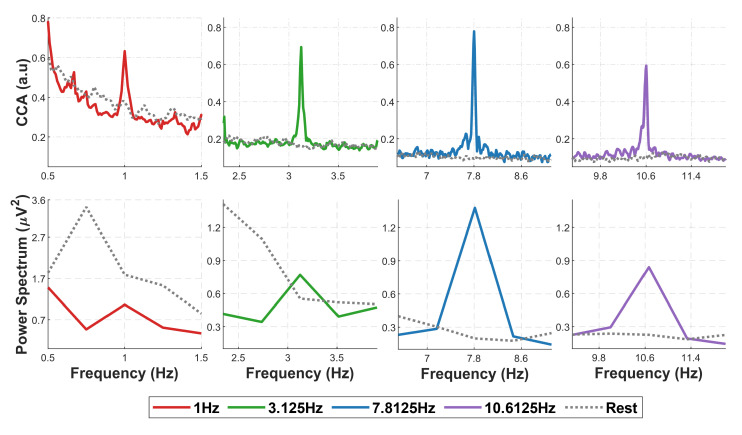
CCA (top row) and Power Spectrum (bottom row) of 20-s-long EEG data segments relative to visual stimulation at (from left to right) 1, 3.125, 7.8125, 10.6125 Hz and on resting state data adjacent to the same segments, from a representative subject.

**Figure 4 sensors-22-09803-f004:**
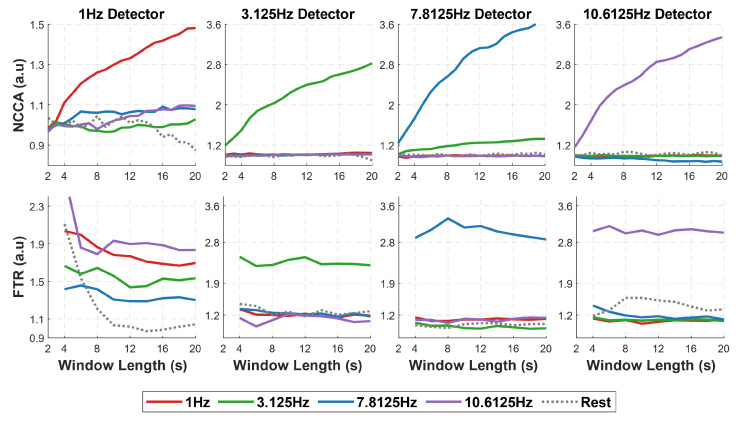
NCCA (top row) and FTR (bottom row) detectors at the four stimulation frequencies (from left to right: 1, 3.125, 7.8125, and 10.6125 Hz) applied to data segments relative to the four stimulation frequencies and to adjacent rest data as a function of the data window length.

**Figure 5 sensors-22-09803-f005:**
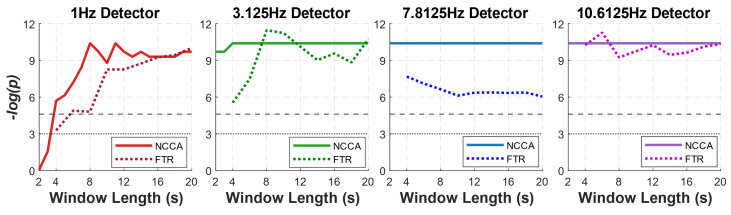
Statistical significance of NCCA and FTR (where *p* is the *p*-value of the one-tailed Wilcoxon signed rank test between the CCA/power spectrum at the stimulation frequency and the CCA/power spectrum of the background EEG estimated from the neighboring frequency bins) at the four stimulation frequencies (from left to right: 1, 3.125, 7.8125, and 10.6125 Hz) applied to data segments relative to the corresponding stimulation frequencies, as a function of data window length.

## Data Availability

The data presented in this study are available on request from the corresponding author.

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
