# Peer review of "Efficient Low-Frequency SSVEP Detection with Wearable EEG Using Normalized Canonical Correlation Analysis"

_sensors, 2022, doi:10.3390/s22249803_

Round 1
Reviewer 1 Report
It was challenging for the authors to use ultra-low frequency stimuli and perform the detection with 8 EEG channels. But there are some issues to be clear about before publication.
(1) In abstract, a frequency-normalized index based on Canonical Correlation Analysis is called NCCA for short? Please add NCCA in the keywords.
(2) The innovation of the manuscript is not outstanding.
(3) Why the authors only selected four frequency points: 1 Hz, 3.125 Hz, 7.8125 Hz and 10.6125 Hz. What is the meaning of incommensurable stimulation rates? Please add more frequency points
(4) Please give the name of the 8 EEG channels.
(5) Only 10 participants were recruited. It is hard to achieve reliable conclusions. Please added samples.
(6) The discussion section needs to be rewritten. Please add research results related to this dataset for comparison.
Reviewer 2 Report
“Efficient Low-Frequency SSVEP detection with wearable EEG using Normalized Canonical Correlation Analysis”(sensors-1996531)
This manuscript aims to test the efficiency of low-frequency SSVEP detection with wearable EEG using Normalized Canonical Correlation Analysis. The results revealed that NCCA correctly and rapidly detects SSVEP at the stimulation frequency within a few cycles of stimulation, even at the
lowest frequency (6 s recordings are sufficient for a stimulation frequency of 1 Hz), outperforming a state-of-the-art normalized power spectral measure. Considered that no preliminary artifact correction or channel selection was required for NCCA, great research and practical implications were ensured. Overall, this topic is interesting and important and the current results are encouraging. However, some concerns appeared after reading the whole manuscript.
1. How did you determine the sample size? Did you calculate the sample size needed before formal study?
References:
Lakens, D. (2022). Sample size justification. Collabra: Psychology, 8(1), 33267.
Larson, M. J., & Carbine, K. A. (2017). Sample size calculations in human electrophysiology (EEG and ERP) studies: A systematic review and recommendations for increased rigor. International Journal of Psychophysiology, 111, 33-41.
Niso, G., Krol, L. R., Combrisson, E., Dubarry, A. S., Elliott, M. A., François, C., ... & Chaumon, M. (2022). Good scientific practice in EEG and MEG research. NeuroImage, 257, 119056.
Pernet, C., Garrido, M. I., Gramfort, A., Maurits, N., Michel, C. M., Pang, E., ... & Puce, A. (2020). Issues and recommendations from the OHBM COBIDAS MEEG committee for reproducible EEG and MEG research. Nature neuroscience, 23(12), 1473-1483.
2. Please add the gender information of the participants.
3. What does the “N” of the “NCCA” stand for in the abstract?
4. What does “SNR” mean? Please explain it when it firstly appeared in the manuscript.
5. The “Institutional Review Board Statement” and “Informed Consent Statement” parts should be included in the manuscript.
Round 2
Reviewer 1 Report
At present version, I have one major concern. The authors only selected four frequency points: 1 Hz, 3.125 Hz, 7.8125 Hz and 10.6125 Hz, and they are unrepresentative. please add more frequency points.
